# Optimizing Bioink Composition for Human Chondrocyte Expression of Lubricin

**DOI:** 10.3390/bioengineering10090997

**Published:** 2023-08-23

**Authors:** Kari Martyniak, Sean Kennedy, Makan Karimzadeh, Maria A. Cruz, Oju Jeon, Eben Alsberg, Thomas J. Kean

**Affiliations:** 1Biionix Cluster, College of Medicine, University of Central Florida, Orlando, FL 32827, USA; 2Department of Biomedical Engineering, University of Illinois Chicago, Chicago, IL 60607, USA; ojeon2@uic.edu (O.J.); ealsberg@uic.edu (E.A.)

**Keywords:** human articular chondrocyte reporter, 3D bioprinting cartilage, surface zone articular cartilage, oxidized methacrylated alginate, lap-shear, lubricin chondrocyte reporter, engineered extracellular matrix reporter, tissue engineered cartilage

## Abstract

The surface zone of articular cartilage is the first area impacted by cartilage defects, commonly resulting in osteoarthritis. Chondrocytes in the surface zone of articular cartilage synthesize and secrete lubricin, a proteoglycan that functions as a lubricant protecting the deeper layers from shear stress. Notably, 3D bioprinting is a tissue engineering technique that uses cells encapsulated in biomaterials to fabricate 3D constructs. Gelatin methacrylate (GelMA) is a frequently used biomaterial for 3D bioprinting cartilage. Oxidized methacrylated alginate (OMA) is a chemically modified alginate designed for its tunable degradation rate and mechanical properties. To determine an optimal combination of GelMA and OMA for lubricin expression, we used our novel high-throughput human articular chondrocyte reporter system. Primary human chondrocytes were transduced with *PRG4* (lubricin) promoter-driven *Gaussia* luciferase, allowing for temporal assessment of lubricin expression. A lubricin expression-driven Design of Experiment screen and subsequent validation identified 14% GelMA/2% OMA for further study. Therefore, DoE optimized 14% GelMA/2% OMA, 14% GelMA control, and 16% GelMA (total solid content control) were 3D bioprinted. The combination of lubricin protein expression and shape retention over the 22 days in culture, successfully determined the 14% GelMA/2%OMA to be the optimal formulation for lubricin secretion. This strategy allows for rapid analysis of the role(s) of biomaterial composition, stiffness or other cell manipulations on lubricin expression by chondrocytes, which may improve therapeutic strategies for cartilage regeneration.

## 1. Introduction

Osteoarthritis (OA) is the most common form of arthritis, negatively impacting millions of individuals each year [1]. OA is characterized by the loss of cartilage. Defects occur in the superficial layer of articular cartilage, and continue to degrade down to the sub-chondral bone [2]. This makes the surface zone of articular cartilage the first area impacted by cartilage defects [3,4,5]. The surface zone of articular cartilage functions to protect the deeper layers from shear stress [2,6]. This layer is in contact with the synovial fluid of the joint, and both the collagen fibers and chondrocytes are oriented parallel to the articulating surface [2,6,7]. There are more chondrocytes in this zone compared to deeper zones, and they primarily synthesize and secrete lubricin [2,6,7]. Lubricin is a proteoglycan, derived from the proteoglycan 4 (*PRG4*) gene, which functions as a boundary lubricant [3,8]. It has been shown to decrease the coefficient of friction [9] and prevent synovial cell and protein adhesion to the cartilage surface [10]. Lubricin is essential for fully functional articular cartilage, and mutations to the *PRG4* gene result in camptodactyly-arthropathy-coxa vara-pericarditis syndrome, a disease resulting in poly-articular OA [8,11]. Treatment with recombinant lubricin or lubricin mimetics has been shown to reduce the damaging effects of surgically induced OA in rats [12,13] and OA in ovariectomized rats [14]. Overall, lubricin is critical for functional joint tissue.

Surgeries like matrix-induced autologous chondrocyte implantation are resurfacing methods commonly used to treat cartilage defects [15]. However, these methods have drawbacks including a lack of donor tissue, inflammation, degradation, and biocompatibility [1]. Notably, 3D bioprinting is a tissue engineering technique involving the simultaneous extrusion of biomaterials and living cells [16,17]. Since the cells are encapsulated within the biomaterials, there is no need for post-fabrication cell seeding and therefore there is no cell penetration limitation [18]. Significantly, 3D bioprinting has the ability to create patient-specific bioactive scaffolds to treat tissue defects and/or to resurface a joint. Biomaterials can be optimized for stimulating lubricin expression, effectively recreating that protection lost upon injury to the tissue. Patient-specific cells can be encapsulated within biomaterials to further improve biocompatibility. Overall, 3D bioprinting can be used to create a scaffold specifically shaped for the defect site, without the drawbacks of traditional therapies.

To 3D bioprint the surface zone for articular cartilage defects, biomaterials should promote lubricin expression. For an ideal construct, biomaterials also need to mimic the extracellular matrix (ECM), fill the defect space, and maintain that space while integrating with the surrounding tissue [19]. For this study, we chose to use gelatin methacrylate (GelMA) in combination with oxidized methacrylated alginate (OMA). GelMA is a well characterized and frequently used biomaterial for 3D bioprinting cartilage [20,21,22,23,24,25,26,27,28,29,30,31,32,33,34,35,36,37,38]. OMA is chemically modified alginate developed for its tunable physical properties, such as degradation rate [39,40]. It has been shown to be ideal for 3D bioprinting due to its shear-thinning ability after calcium crosslinking [41]. Once shear stress is removed, OMA recovers its mechanical properties rapidly [41]. Oxidation of alginate prior to methacrylation alters the uronate residue conformations, making it more vulnerable to hydrolysis and provides enhanced tuning of degradation rate [40]. The methacrylation of both biomaterials enables photocrosslinking by visible light when combined with the photoinitiator lithium phenyl-2,4,6-trimethylbenzoylphosphinate (LAP) [42] for further stability and enhanced mechanical properties.

In this study, we developed a novel reporter system with a *PRG4* promoter-driven *Gaussia* luciferase (HuPRG4gLuc) in primary human chondrocytes. Upon *PRG4* stimulation, *Gaussia* luciferase is secreted from the cells, making this a non-destructive assay [25,43,44]. While this is the first study using the HuPRG4gLuc cells, we have previously used a similar reporter system to analyze type II collagen expression. The type II collagen reporter cells have been used for micronutrient optimization in chondrogenic media for a murine chondrogenic cell line (ATDC5s) [43], primary rabbit chondrocytes [44], and for biomaterial optimization with human articular chondrocytes [25]. In this study, we expand upon previous biomaterial optimization by focusing on new biomaterials and how they impact lubricin expression in human articular chondrocytes.

To streamline this biomaterial optimization, we also used a Design of Experiment (DoE) approach. DoE software (Design-Expert, Stat-Ease) uses statistical modeling to generate a set of combinations, while also limiting the number of groups tested [45,46]. Utilizing this method effectively decreases the amount of time and materials used to determine optimal groups for 3D bioprinting [25]. Using biomaterial combinations generated by the DoE screen with the HuPRG4gLuc cells, we developed a streamlined approach to identify biomaterial combinations that increase lubricin expression. This systematic approach identified an optimal combination of GelMA and OMA based on lubricin promoter-driven luminescence, as well as on biochemical and mechanical data (Figure 1A).

## 2. Materials and Methods

### 2.1. Engineering of Lubricin Promoter-Driven Reporter Human Chondrocytes (HuPRG4gLuc)

Human articular chondrocytes (64-year-old, non-diabetic female) were isolated as previously described [25]. Briefly, macroscopically normal cartilage tissue was dissected from discarded surgical tissue from a patient undergoing total joint replacement. Cartilage was diced <1 mm^3^ and sequentially digested in hyaluronidase, followed by collagenase. Cells were resuspended (95% FBS, 5% DMSO), frozen, and stored in liquid nitrogen. Lentiviral vector plasmids psPAX2 (plasmid #12260; Addgene, Watertown, MA, USA), pMD2G (plasmid #12259;), and PRG4-gLuc (9394 Bp, Genecopoeia, Rockville, MD, USA, Figure 1B) were purified from transformed competent *E. coli* (GCI-L3; Genecopoeia) using a commercial kit (ZymoPURE II Plasmid Maxiprep Kit; Zymo Research Corp., Irvine, CA, USA). HEK293Ta cells (Genecopoeia) were transfected with the purified plasmids using calcium phosphate nanoparticles. Lentiviral particles were collected from the media and concentrated via ultracentrifugation (30,000 RCF, 8 h, 4 °C). Human articular chondrocytes were thawed from stocks of uncultured cells, seeded, and grown prior to lentiviral infection. Chondrocytes were incubated with pseudolentiviral particles (MOI 2.3) at 4 °C for 20 min and then moved to a cell culture incubator (humidified 37 °C atmospheric oxygen, 5% CO_2_) for 11 h. The cells were grown to ~90% confluence and then passaged onto synoviocyte matrix-coated flasks and purified with puromycin selection (2 µg/mL) for 7 days [25,47]. The remaining cells were grown to ~90% confluence prior to being trypsinized, neutralized, and cryopreserved with FBS (95%) and DMSO (5%).

### 2.2. HuPRG4gLuc Reporter Cell Aggregate Culture and TGFβ1 Dose Response

HuPRG4gLuc reporter chondrocytes were thawed from frozen stocks and seeded onto synoviocyte-derived matrix-coated flasks to expand to confluence (~5 days, 37 °C, 5% CO_2_ and 5% O_2_) [25,47,48]. Upon confluence, cells were trypsinized (0.25% Trypsin/EDTA, 5 min, 37 °C), neutralized with growth media (DMEM/F12 (Gibco, Carlsbad, CA, USA), supplemented with 10% FBS (mesenchymal stromal cell selected [49] and 1% penicillin-streptomycin (Hyclone, Marlborough, MA, USA)), and then centrifuged (5 min, 500 RCF, room temperature [RT]). Cells were then resuspended in defined chondrogenic media (DMEM-HG 93.24% (Lonza, Walkersville, MD, USA), 1% 10^−5^M dexamethasone (Sigma, Burlington, MA, USA), 1% ITS+premix (Becton-Dickinson, Franklin Lakes, NJ, USA), 1% Glutamax (Hyclone), 1% 100 mM Sodium Pyruvate (Hyclone), 1% MEM Non-Essential Amino Acids (Hyclone), 0.26% 50 mM L-Ascorbic Acid Phosphate (Wako, Richmon, VA, USA), 0.5% Fungizone (Life Technologies, Carlsbad, CA, USA)) [25,43,50], seeded (50,000 cells/well) onto a non-adherent 96-well plate (Greiner Bio-One, Monroe, NC, USA), and then centrifuged to form cell aggregates (5 min, 500 RCF, RT). Cell aggregates were cultured for 3 weeks in defined chondrogenic media containing TGFβ1 (0–40 ng/mL, Peprotech, Cranbury, NJ, USA), and fed 3 times a week (37 °C, 5% CO_2_ and 5% O_2_). Media containing the secreted *Gaussia* luciferase was sampled twice a week for luminescence (Section 2.3). On day 22, cell aggregates were divided between histological analysis (Section 2.4), biochemical assessment (Section 2.5), or qPCR gene expression analysis (Section 2.6).

### 2.3. Luciferase Assessment

As previously described, during the 22-day culture period, luminescence was assessed twice a week [25]. Culture media (20 µL/well) was combined with a stabilized luciferase reaction mix (50 µL/well) on a white, 96-well plate (Greiner Bio-One). The final concentration of each component of the stabilized reaction mix was 0.09 M MES, 0.15 M ascorbic acid, and 4.2 µM coelenterazine. Luminescence was read on a plate reader (PerkinElmer EnVision 2104 Multilabel Reader, Waltham, MA, USA).

### 2.4. Histology

Samples were fixed in 10% neutral buffered formalin (Epredia, Kalamazoo, MI, USA) for ~2 days, dehydrated, and then embedded in paraffin wax. To assess sulfated GAG content, rehydrated samples were stained with safranin-O (Electron Microscopy Sciences, Hatfield, PA, USA) and counterstained with fast green (Alfa Aesar, Ward Hill, MA, USA) and Weigert’s iron hematoxylin (Electron Microscopy Sciences). Immunohistochemistry was performed for type II collagen content, as previously described [25], and for lubricin. Briefly, for type II collagen, antigen retrieval was done with Pronase (1 mg/mL for 10 min at RT, Sigma) followed by blocking with BSA (3% *w*/*v*, Cohn Fraction V, Alfa Aesar). Primary antibody (1:200 in 1% BSA, DSHB, II-II6B3) incubation was for 2 h at RT and secondary antibody (1:2000 in 1% BSA, Biotinylated horse anti-mouse, Vector Labs, Newark, CA, USA) incubation was for 1 h at RT. Finally, HRP-conjugated streptavidin (1:5000 in 1% BSA, Invitrogen, Carlsbad, CA, USA) was incubated for 30 min at RT and Vector VIP Peroxidase Substrate Kit (Vector Labs) was used to develop staining. For lubricin, antigen retrieval was performed with hyaluronidase (10 mg/mL in 20 mM sodium acetate, Sigma) incubation at 37 °C for 30 min. Samples were blocked with 3% BSA. Primary lubricin antibody (1:400, 1% BSA, Millipore MABT401, Burlington, MA, USA) was incubated for 90 min at RT, and secondary antibody was incubated for 30 min at RT. HRP-streptavidin incubation and stain development was the same as for type II collagen. All sections were counter stained with Fast Green (VWR) and imaged (Keyence BZ-X810 microscope, Itasca, IL, USA).

### 2.5. Biochemical Assays

As previously described [25], frozen samples were digested in papain (0.025 mg/mL papain in 50 mM sodium phosphate, 2 mM EDTA, 2 mM cysteine) [25,50,51] overnight. Hoechst dye was used to quantify the DNA content of the digest, with calf thymus DNA (Sigma) as the standard. Samples were read on a plate reader (Ex365, Em460nm, SpectraMax iD5, Molecular Devices, San Jose, CA, USA). Dimethylmethylene blue (5 mL of 3.2 mg/mL DMMB dissolved in 200 proof ethanol added to 1 L of 40 mM glycine, 40 mM NaCl at pH 1.5) colorimetric assay was used to quantify the GAG content of the digest with absorbance readings at 525 nm and 595 nm (correction). Hydroxyproline (HDP) content was measured for total collagen quantification. Papain-digested samples were acid-hydrolyzed overnight (10:1 vol/vol, 6 M HCl, 110 °C) and then dried overnight (70 °C). Copper sulfate (0.15 M) and NaOH (2.5 M) were added and incubated (50 °C, 5 min). Oxidation occurred through 10-min incubation with hydrogen peroxide (6% H_2_O_2_, 50 °C). Sulfuric acid was added (1.5 M H_2_SO_4_) prior to reaction with Ehrlich’s reagent (10% *w*/*v* 4-dimethylamino benzaldehyde in 60% isopropanol, 26% perchloric acid, 14% MΩ water) for a final incubation (70 °C, 16 min). Absorbance was read at 505 nm (SpectraMax iD5, Molecular Devices) with the hydroxyproline standard curve generated using hydroxyproline (Sigma). For the biomaterials, samples without cells were used to subtract the background generated by GelMA or OMA.

### 2.6. HuPRG4gLuc Reporter Cells qPCR Assessment

HuPRG4gLuc reporter chondrocytes were seeded in cell aggregate culture (Section 2.3) and cultured in defined chondrogenic media supplemented with 0–1 ng/mL TGFβ1. On day 22, cell aggregates were frozen (−80 °C) in an RNA lysis buffer for subsequent RNA extraction (Invitrogen RNA Purelink minikit). mRNA was isolated using column purification with on-column DNase treatment. RNA ScreenTape (Agilent Technologies, Santa Clara, CA, USA) was used to assess RNA purity and integrity. cDNA was synthesized (Maxima H cDNA kit) and qPCR was performed (QuantStudio 7 flex, Applied Biosystems, Waltham, MA, USA). The following primers were used for gene expression analysis: Hypoxanthine Phosphoribosyltransferase 1 (*HRPT*, reference gene, forward primer: 5′ ATTGACACTGGCAAAACAATGC 3′, reverse primer: 5′ TCCAACACTTCGTGGGGTCC 3′, [25,47]), *Gaussia* luciferase (gLuc, forward primer: 5′ ACGCTGCCACACCTACGA 3′, reverse primer: 5′ CCTTGAACCCAGGAATCTCAG 3′ [25,43]), and lubricin (*PRG4*, forward primer: 5′ TTGCTCCTCTCTGTTTTCGT 3′, reverse primer: 5′ ATACCCTTCCCCACATCTCCC 3′). A mixture of the primers, SYBR green (Applied Biosystems, Thermofisher Scientific), and cDNA was run with cycling parameters: 95 °C for 20 s then 45 cycles of 95 °C 10 s, 60 °C 20 s, 72 °C 19 s, followed by melt curve analysis. CT values were normalized to *HRPT* expression and gLuc vs. *PRG4* relative gene expression was plotted with 95% confidence bands (GraphPad Prism, Boston, MA, USA).

### 2.7. Design of Experiment (DoE) Screen Design

Stat-Ease, Design-Expert (Version 13; Minneapolis, MN, USA) was used to generate testing conditions for screening combinations of GelMA and OMA at different photocrosslinking times. A mixture model was used in an optimal combined design generating a total of 60 conditions. Constraints were set so the final percentage of GelMA in the mixture was between 0% and 12%, while OMA percentage was between 0% and 2%, with the rest of the mix being PBS, between 86% and 98%. The sum of the final percentages of GelMA and OMA was set to be between 2% and 14%, and the sum of GelMA, OMA, and PBS would always equal 100% (Appendix A). Crosslinking time was the numeric factor, ranging between 15 s and 60 s. To determine if the addition of calcium prior to cell encapsulation had an impact on lubricin expression, calcium inclusion was a categoric factor, resulting in groups with or without calcium (final concentration 1.8 mM CaCl_2_). DoE-generated testing conditions were combined with HuPRG4gLuc cells (Section 2.9), and lubricin promoter-driven luciferase expression was assessed (Section 2.3). Luminescence data was input into the DoE software where it suggested a square root transformation to fit the model. ANOVA analysis was used to identify if the biomaterial mixture, calcium chloride addition, or crosslinking time had a statistically significant impact on lubricin-driven luminescence.

### 2.8. Oxidized Methacrylated Alginate (OMA) Synthesis and Characterization

Oxidized alginate was prepared by reacting sodium alginate (LF120M, 1% aqueous solution viscosity = 251 mPa, FMC Biopolymer, Philadelphia, PA, USA) with sodium periodate (Sigma) [39]. Briefly, sodium alginate (10 g) was dissolved in ultrapure deionized water (diH_2_O, 900 mL) overnight. Sodium periodate (0.54 g) was dissolved in 100 mL diH_2_O, added to alginate solution under vigorous stirring to achieve 5% theoretical alginate oxidation, and allowed to react in the dark at room temperature for 24 h. Methacrylation (30% theoretical) was performed to obtain OMA (5OX30MA) by reacting oxidized alginate with 2-aminoethyl methacrylate hydrochloride (AEMA, Polysciences, Warrington, PA, USA). To synthesize OMA, 2-morpholinoethansulfonic acid (19.52 g, Sigma) and NaCl (17.53 g, Fisher Scientific, Waltham, MA, USA) were directly added to the oxidized alginate solution and the pH was adjusted to 6.5. N-hydroxysuccinimide (NHS, 1.764 g, Fisher Scientific) and 1-ethyl-3-(3-dimethylaminopropyl)-carbodiimide hydrochloride (EDC, 5.832 g, Oakwood Chemical, Estill, SC, USA) were added to the solution under stirring to activate 30% of the carboxylic acids of the oxidized alginate. After 5 min, AEMA (2.532 g, molar ratio of NHS:EDC:AEMA = 1:2:1) was added to the solution and the reaction was maintained in the dark at room temperature for 24 h. The reaction mixture was precipitated into excess acetone, dried in a fume hood, and rehydrated to a 1% *w*/*v* solution in diH_2_O for further purification. The OMA was purified by dialysis against diH_2_O using a dialysis membrane (MWCO 3500 Da, Spectrum Laboratories, New Brunswick, NJ, USA) for 3 days, treated with activated carbon (5 g/L, 100 mesh, Oakwood Chemicals) for 30 min, filtered (0.22 µm filter), and lyophilized. To determine the levels of alginate methacrylation, the OMA was dissolved in deuterium oxide (2% *w*/*v*), and ^1^H-NMR spectrum was recorded on an NMR spectrometer (600 MHz, Bruker) using 3-(trimethylsilyl)propionic acid-d4-sodium salt (0.05% *w*/*v*) as an internal standard.

### 2.9. Preparation of GelMA and OMA Stocks and Combinations

GelMA (58% methacrylated, 167 kDa, Rousselot, Ghent, Belgium) and OMA were reconstituted in PBS containing 0.05% lithium phenyl-2,4,6-trimethylbenzoylphosphinate (LAP, Cellink, Carlsbad, CA, USA). Stocks were made by combining the material with PBS/LAP in 1.6 mL microtubes on a tube warmer at 50 °C and shaken (800 rpm) until fully dissolved. To prepare biomaterial combinations, calculated amounts of each stock were added to 1.6 mL microtubes on the tube warmer at 50 °C, with a quick vortex to fully combine. PBS containing 0.05% LAP was added if further dilutions were necessary. All stocks and combinations were made less than 24 h prior to use and stored in the dark at 4 °C. To encapsulate cells, biomaterial combinations were warmed on the tube warmer to 37 °C with cells added and vortexed to mix. The cell-biomaterial mixtures were either pipetted onto 96 well white plates with a clear bottom for the DoE screen (Section 2.7) and subsequent validation or added to barrels for 3D bioprinting (Section 2.10).

### 2.10. 3D Bioprinting

All 3D bioprinting was performed using a BioAssemblyBot pneumatic extrusion 3D bioprinter (Advanced Solutions, Louisville, KY, USA) and constructs designed using the Tissue Structure Information Modeling (TSIM) software. Biomaterials were printed using 3 mL UV blocking amber barrels (Nordson, Westlake, OH, USA), 25G SmoothFlow tapered tips (Nordson), and the print settings shown in Table 1 and Figure 2. The hot tool attachment was adapted for use with 3 mL barrels to warm biomaterials. After printing, constructs were photo-crosslinked (Luck Laser, 405 nm, 300 mW) with the light focused to an 8 mm beam diameter and set 3.5 cm above the construct. Cylindrical constructs (8 mm diameter × 1 mm height) were bioprinted containing HuPRG4gLuc cells for luminescence assay, mechanical characterization, histology, and biochemical assay. To quantify cell viability, single layer rectangular cuboids (2 mm × 6 mm × 0.3 mm) were bioprinted containing HuPRG4gLuc cells.

All printed with 25TT (25-gauge Taper Tip needle, Nordson), stage temp of 35 °C, line width 0.85 and height of 0.2 mm.

### 2.11. Cell Viability

Cell viability was assessed using live/dead staining on days 0, 1, and 7. Groups 14% GelMA, 16% GelMA or 14% GelMA, and 2% OMA containing HuPRG4gLuc cells (1 million cells/mL final concentration) were 3D bioprinted as previously described (Section 2.10) or pipetted (control) in triplicate for each time point. As previously described [25], at each time point the staining solution (calcein-AM (2 µM, Invitrogen) and ethidium homodimer-1 (4 µM, Invitrogen) in sterile PBS) was added to each well, incubated for 25 min (37 °C, 5% CO_2_), and then removed and replaced with PBS. Images were taken using a Pico Imager (Molecular Devices) and then processed and analyzed using ImageJ/Fiji with the Stardist plugin [52] with >100 cells quantified for each sample.

### 2.12. Mechanical Characterization

#### 2.12.1. Dynamic Mechanical Analysis (DMA) 

Rheological testing was performed (DMA 242E Artemis, Netzsch, Burlington, MA, USA) with a set strain of 10% (100 µm) and frequencies of 0.1, 1, and 5 Hz at room temperature. Disc width and height were measured with calipers (Netzsch). Each disc was also imaged, and surface area quantified using ImageJ/Fiji. Discs were maintained in PBS for the duration of the test. The storage modulus (E′), loss modulus (E″), and tan delta were measured.

#### 2.12.2. Lap-Shear

Static coefficient of friction and kinetic coefficient of friction were determined through a lap shear test on disc constructs. Day 0 discs were cast in an 8 mm diameter, 1 mm high silicone mold, while day 22 constructs were 3D bioprinted discs of the same dimensions containing HuPRG4gLuc cells and cultured for 22 days. Constructs were frozen (−80 °C) until use. Lap shear testing was performed using a TA.XTplusC texture analyzer (Stable Micro Systems, Hamilton, MA, USA; Appendix A). The samples were adhered to a microscope slide (VWR) and placed securely into the top tensile clamp. A second microscope slide was secured into the bottom tensile clamp. The sample was fully submerged in PBS for the duration of the test. The sample was aligned until it was touching the second microscope slide and then compressed by ~200 µm, creating a normal force of 1.62 N. Normal force was determined by a force sensitive resistor (DF9–40, Yosoo Health Gear). The kinetic force was determined by the peaks of the graph generated from a shear sine wave test. The coefficient of friction was calculated from the force generated on the graphs, divided by the normal force.

### 2.13. Swelling and Degradation

For this experiment, 3D bioprinted discs (8 mm diameter × 1 mm height) were frozen (−80 °C) in pre-weighed 1.6 mL microtubes (W_t_), lyophilized, and initial dry weights (W_i_) measured. Chondrogenic media (700 µL, Section 2.2) was added to each tube, fully submerging the lyophilized discs, and then incubated (37 °C, 5% CO_2_) for set time points. Media was changed weekly. On days 1, 11, and 22, all media was removed, and swollen (W_s_) weights were measured. After weighing, samples were frozen (−80 °C), lyophilized again, and then weighed (W_d_). To calculate the swelling ratio (Q), the swollen weight was divided by the initial weight (Q = (W_s_ − W_t_)/(W_d_ − W_t_)). The percent mass loss was calculated by ((W_i_ − W_t_) − (W_d_ − W_t_))/(W_i_ − W_t_) × 100.

### 2.14. Lubricin (PRG4) ELISA

To quantify secreted lubricin, cell culture medium was collected from the TGFβ1 dose response on day 16 and from the 3D bioprinted disc constructs on days 1, 10, and 22. Cell culture medium was frozen at −20 °C until use. Enzyme-Linked Immunosorbent Assay (ELISA) kit (DuoSet ELISA Ancillary Reagent Kit and Human Lubricin/PRG4 kit, R&D systems) was used following the manufacturer’s protocol. Lubricin content was calculated based on the standard curve.

### 2.15. Statistical Analysis

For the DoE screen, design of experiment analysis was completed as previously described (Section 2.7). For all other experiments, statistical analysis was completed using GraphPad Prism (Version 9.0). All experiments had 3–9 replicates (*n* = 3–9) and results are shown ± standard deviation. A *p* value of <0.05 was considered statistically significant.

## 3. Results

### 3.1. Stimulation of Lubricin by TGFβ1 in Human Primary Articular Chondrocytes

To characterize human primary articular chondrocytes engineered with a PRG4 promotor-driven Gaussia luciferase (HuPRG4gLuc), cell aggregates were cultured with varying doses of TGFβ1, a known inducer of lubricin [3,7,53]. Luminescence, a proxy for lubricin, increased in a TGFβ1 dose-dependent manner, as shown by the response curves for days 10 and 22 in Figure 3A. The excitatory concentration producing a half-maximal response (EC50) was 2.5 ng/mL on day 10 and 4.2 ng/mL on day 22 (Figure 3A). Lubricin protein secretion also had a TGFβ1 dose-dependent increase, with an EC50 of 4.3 ng/mL (Figure 3B), very similar to the luminescence EC50. This is further supported by qPCR gene expression analysis which correlated PRG4 expression with gLuc gene expression (Figure 3C). There was a TGFβ1 dose-dependent increase of both DNA (EC50 = 2.8 ng/mL, Figure 3D) and GAG/DNA (EC50 = 4.8 ng/mL, Figure 3E), which is supported by Safranin-O staining (Figure 4). While hydroxyproline (HDP) per DNA (Figure 3F) was consistent across groups, immunohistochemical staining for type II collagen showed an increase in staining intensity correlating with an increase in TGFβ1 (Figure 4). Lubricin immunostaining also showed an increase in staining intensity as TGFβ1 concentration increased (Figure 4). Together these results support using luminescence as a proxy for lubricin expression and confirmed that the engineered cells retained chondrogenic capacity.

### 3.2. Characterization of OMA

To prepare the biodegradable and photocrosslinkable OMA, sodium alginate was oxidized using sodium periodate in an aqueous solution at room temperature for 24 h, and then methacryloyl groups were introduced onto the oxidized alginate main chains as shown in Figure 5. The actual oxidation and methacrylation degrees of alginate were calculated from ^1^H-NMR spectra, which are shown in Figure 5. In the ^1^H-NMR spectrum of the oxidized alginate, as a result of the oxidation, the new proton (Figure 5, a’) on carbon 1 was formed at 5.5 ppm. The ^1^H-NMR spectrum of the OMA exhibited newly formed proton peaks of vinyl methylene and methyl by the reaction with AEMA at 6.2, 5.6, and 1.9 ppm, respectively. The actual oxidation of the hydroxyl groups on carbons 2 and 3 of the repeating units of the sodium alginate was 4.8% (5% theoretical) and the actual methacrylation of the repeating units of the oxidized alginate was 18.3% (30% theoretical).

### 3.3. Design of Experiment (DoE) Screen of GelMA and OMA Combinations for Lubricin Expression

To identify the optimal combinations of OMA and GelMA for lubricin expression, HuPRG4gLuc cells were mixed with biomaterial combinations generated by the DoE. Sixty different combinations, at different crosslinking times, with or without the addition of calcium chloride, were tested. At all timepoints the mixture of the biomaterials had a significant impact on luminescence (*p* < 0.001, ANOVA (Appendix A)), while crosslinking time and calcium chloride were not significant factors. Day 22 luminescence is shown on 3D surface plots in Figure 6. Groups that contained only OMA had the lowest luminescence (Figure 6A). Increasing GelMA to 6% increased luminescence as compared to OMA alone (Figure 6B), with the shortest crosslinking time (15 s) and no OMA having the highest luminescence. Luminescence further increased by increasing GelMA concentration to 12% (Figure 6C). In the groups that had 12% GelMA (Figure 6C), luminescence further increased as the final percentage of OMA increased, with the highest luminescence expression in 12% GelMA/2% OMA. Data was normally distributed as shown in Figure 6D.

### 3.4. Validation of the DoE Screen

Validation of the DoE screen results was performed using the HuPRG4gLuc cells cultured in combinations of GelMA and OMA. All selected biomaterial groups had higher luminescence than the cell aggregate control (Figure 7A), indicating the mixing of chondrocytes in biomaterials increased lubricin expression. Day 22 luminescence data is shown in Figure 7B–E. To investigate the lack of effect of crosslinking time seen in the screen, the optimal group from the screen (12% GelMA/2% OMA) was retested at the three crosslinking times (15 s, 38 s, and 60 s). Crosslinking time did not have a significant effect on luminescence (Figure 7B). Since the highest final percentage of both GelMA and OMA had the highest luminescence in the screen, both were further increased, and luminescence was assessed. Increasing the final OMA percentage to 4% did not positively impact luminescence at either crosslinking time (Figure 7C). Increasing the final GelMA percentage to 14%, while keeping OMA consistent at 2%, did significantly increase luminescence as compared to 12% GelMA/2% OMA at both crosslinking times (Figure 7D). To determine if this increase in luminescence was solely due to increasing the GelMA percentage, 14% GelMA alone was compared to 14% GelMA/2% OMA at both crosslinking times (Figure 7E). The group containing 2% OMA had significantly higher luminescence as compared to 14% GelMA alone at 15 s crosslinking, but not 38 s (Figure 7E). The 14% GelMA/2% OMA group after 15 s crosslinking was consistently the group with the highest luminescence starting on day 10 (Figure 7A). DNA content stayed consistent across all 14% GelMA groups (Appendix A), but GAG/DNA was significantly higher in 14% GelMA/2% OMA group at both crosslinking times (Appendix A). Based on the luminescence, DNA and GAG data, the 14% GelMA/2% OMA at 15 s was determined to be optimal for lubricin expression.

### 3.5. Cell Viability

Cell viability was assessed in groups 14% GelMA, 14% GelMA/2% OMA, and 16% GelMA to ensure the bioprinting process and biomaterials were biocompatible (Figure 8). Groups were either 3D bioprinted or pipetted and viability was quantified on days 0, 1, and 7. On days 0 and 1 cell viability was significantly decreased in all 3D bioprinted groups, as compared to their respective pipetted controls (Appendix A). By day 7, the 14% GelMA and 16% GelMA printed groups still had significantly lower cell viability as compared to their pipetted controls, but the 14% GelMA/2% OMA printed group was significantly higher than the pipetted (Appendix A). The 14% GelMA printed group cell viability significantly decreased over 7 days from 77% to 61% (Figure 8). The 14% GelMA/2% OMA group stayed consistent across all 7 days at around 72% viability (Figure 8). Finally, on day 0, the 16% GelMA printed group had the lowest viability at only 54%, but it significantly increased to 72% by day 7 (Figure 8).

### 3.6. 3D Bioprinted Disc Construct Containing HuPRG4gLuc Cells

The 14% GelMA/2% OMA after 15 s crosslinking had the highest luminescence in the validation, and improved GAG deposition; therefore, it was used for 3D bioprinting disc constructs containing HuPRG4gLuc cells. Controls: 14% GelMA, as a GelMA only control, and 16% GelMA, as a total solid content control, both with 15 s crosslinking were also printed into discs. Luminescence was assessed over 22 days. The 16% GelMA group had significantly higher luminescence starting on day 3 as compared to the other two groups (Figure 9A). Day 10 and Day 22 had the same trend with 16% GelMA having significantly higher luminescence, and no difference between the 14% GelMA/2% OMA and 14% GelMA groups (Figure 9B).

DNA and GAG content were quantified from constructs on day 0 and day 22. DNA content stayed consistent in both 14% and 16% GelMA, but the 14% GelMA/2% OMA had a significant increase from day 0 to day 22, as well as having significantly more DNA than both the 14% and 16% GelMA groups on day 22 (Figure 10A). The 14% GelMA/2% OMA and 16% GelMA had a significant increase in GAG from day 0 to day 22, while 14% GelMA did not (Figure 10B). The 14% GelMA/2% OMA on day 22 had significantly more GAG than both other groups (Figure 10B). GAG/DNA significantly increased for all three groups by day 22 and the 14% GelMA/2% OMA had significantly more GAG/DNA than the 14% GelMA group (Figure 10C), which was consistent with the GAG/DNA from the validation (Appendix A).

Secreted lubricin content was quantified by ELISA from culture media on days 1, 10, and 22. Lubricin concentration increased from day 1 to day 10, reflecting the luminescence results (Figure 10D). On day 10, 14% GelMA/2% OMA and 16% GelMA had significantly more lubricin than the 14% GelMA group (Figure 10D). On day 22, the 14% GelMA/2% OMA and 16% GelMA groups still had significantly more lubricin than the 14% GelMA group, but 14% GelMA/2% OMA also had significantly more than 16% GelMA (Figure 10D). The 16% GelMA group had a significant decrease in secreted lubricin content from day 10 to day 22 (Figure 10D).

Lubricin content retained in the biomaterials and type II collagen content was assessed by immunohistochemistry (Figure 10E). The 16% GelMA group had more lubricin staining than either of the other groups, while 14% GelMA had very minimal staining (Figure 10E). The 14% GelMA group also had less type II collagen staining (Figure 10E). The type II collagen staining in 16% GelMA was darker, but more pericellular as compared to the 14% GelMA/2% OMA group where it is lighter but more spread out into the material (Figure 10E).

### 3.7. Mechanical Characterization of Disc Constructs

To determine if hydrogel mechanical properties had an impact on lubricin expression, discs were cast and tested on a dynamic mechanic analyzer (DMA). The moduli generated were plotted vs. cumulative luminescence from the validation. There was no trend for either storage modulus or loss modulus (Appendix A) with the R^2^ values being 0.2670 and 0.1319, respectively. This data indicates that within this range of the biomaterial mechanical properties (E’ 10–50 kPa), viscoelasticity played a minimal role in lubricin expression.

DMA analysis was also carried out to characterize 3D bioprinted discs containing HuPRG4gLuc cells on day 0 and 22 for biomaterials after 15 s crosslinking. On day 0, both 14% GelMA and 14% GelMA/2% OMA had a storage modulus of ~30 kPa (Figure 11A). The storage modulus for 16% GelMA storage modulus was significantly higher than the other groups at ~60 kPa vs. ~30 kPa (Figure 11A). By day 22, the 14% GelMA group had contracted too much to be reliably tested. The 16% GelMA storage modulus significantly decreased by day 22 to ~24 kPa (Figure 11A). While the 14% GelMA/2% OMA storage modulus decreased to ~12 kPa, but this change was not statistically significant (Figure 11A). These trends were consistent for the loss modulus, tan delta, and complex modulus (Appendix A).

Lap shear testing was completed on both day 0 and day 22 constructs to determine the coefficient of friction. Day 0 kinetic (Figure 11B) coefficient of friction had no significant difference between the groups. As with the DMA testing, 14% GelMA was too small and thin by day 22 to be reliably tested. By day 22, both the 14% GelMA/2% OMA and 16% GelMA groups had a significant decrease in the kinetic coefficient of friction to ~0.03 (Figure 11B). There was a significant difference in the coefficient of friction on day 22 between the 14% GelMA/2% OMA (0.01) and 16% GelMA groups (0.005, Figure 11B).

### 3.8. Shape Fidelity of the Bioprinted Constructs and Degradation of Bioinks

Over the course of the 22 days in culture, the 14% and 16% GelMA groups noticeably decreased in size, while the 14% GelMA/2% OMA retained its shape. To quantify the size, images were taken on day 22 (Figure 12A) and the surface area was measured. Both 14% and 16% GelMA had a significantly smaller surface area as compared to the 14% GelMA/2% OMA group (Figure 12B). To determine the swelling ratio and mass loss, discs were 3D printed without cells and incubated with chondrogenic media. The swelling ratio stayed consistent for all groups over the 22 days (Figure 12C). All constructs lost about ~12% of their mass on day 1 and ~25% by day 22 (Figure 12D). No group lost significantly more than either other group at any time point, when no cells were present.

## 4. Discussion

To study potential treatments for osteoarthritis (OA), we have developed a high-throughput reporter cell system for lubricin expression. Human primary articular chondrocytes were successfully transduced with a lubricin (*PRG4*) promoter-driven *Gaussia* luciferase (HuPRG4gLuc) while retaining chondrogenic capacity. Lubricin expression correlated well with luciferase expression both at the gene and protein level. *Gaussia* luciferase is secreted from the cells allowing for non-destructive, temporal analysis of lubricin expression. In this study, we expand upon our previous biomaterial optimization [25], now focusing on the surface zone of articular cartilage. Lubricin is an essential proteoglycan for articular cartilage function and treatment with lubricin mimetics has been shown to reduce the damage of PTOA in rats [12,13,14]. The integrated use of DoE with a rapid and easy reporter system allowed us to identify an optimal biomaterial composition for 3D bioprinting, stimulating lubricin expression while maintaining the printed shape. To the best of our knowledge, this is the first study to assess lubricin expression in 3D bioprinted cartilage constructs. Potential uses of this technology include the identification of other lubricin stimulating conditions and the utilization of those compositions for 3D bioprinted resurfacing of the joints.

In this study, we used gelatin methacrylate (GelMA) and oxidized methacrylated alginate (OMA). GelMA is a commonly used biomaterial for 3D bioprinting cartilage [20,21,22,23,24,25,26,27,28,29,30,31,32,33,34,35,36,37,38], adopted in about 35% of cartilage 3D bioprinting papers since 2012 [54]. While it has advantageous properties, we sought to optimize its lubricin promoting capacity by mixing with other biomaterials. Ionically crosslinked alginate is another frequently used biomaterial, but in its natural form has the distinct drawback of very slow degradation in humans [40]. One method to accelerate the degradation rate of alginate is oxidation, making it more vulnerable to hydrolysis [40]. To generate a biomaterial with tunable degradation rates, oxidized alginate was methacrylated, thus adding a site for photocrosslinking [40]. Previously, OMA containing human bone marrow-derived mesenchymal stem cells has been 3D bioprinted in complex geometries with high resolution and high cell viability [41]. Our study is the first-time mixtures of GelMA and OMA were 3D bioprinted with primary human articular chondrocytes. Previously, we hypothesized that an imbalance in ECM production and material degradation rate contributed to a decrease in material storage modulus overtime [25]. By combining GelMA and OMA, we believed this would improve the stability of the construct, while still providing the biochemical cues necessary for lubricin expression.

To identify combinations of GelMA and OMA for increased lubricin expression, an initial screen was implemented using groups generated by a Design of Experiment (DoE) approach. This method uses statistical modeling to reduce the number of conditions tested, while still giving a good overview of the design space. The results of this screen showed the highest final percentage of GelMA and OMA, 12% and 2% respectively, had the highest lubricin expression. This was thought to be a combination of the effect of hydrogel stiffness and composition, as was found with type II collagen expression [25]. To investigate whether maximal stimulation had been achieved, the final percentages were further increased. It was found that increasing the final percentage of OMA from 2% to 4% (with 12% GelMA) was not beneficial for lubricin expression; however, increasing GelMA from 12% to 14% was beneficial. In the screen, 12% GelMA/2% OMA had higher lubricin expression as compared to 12% GelMA alone, and in the validation 14% GelMA/2% OMA had higher lubricin expression as compared to 14% GelMA alone in the final week. This indicates that while increasing GelMA increased lubricin expression, the addition of OMA further improved expression. While further increases in either GelMA or OMA may have improved either lubricin-driven luminescence or stability, they make the hydrogel much more difficult to work with.

Based on luminescence and GAG/DNA results of the validation, the 14% GelMA/2% OMA after 15 s crosslinking was chosen for subsequent bioprinting, while 14% GelMA and 16% GelMA (also at 15 s crosslinking) were 3D bioprinted as controls. Luminescence data showed that raising the final GelMA percentage increased the lubricin expression, since the 16% GelMA group had the highest luminescence. However, 14% GelMA/2% OMA secreted more lubricin from the construct as shown by ELISA, while the 16% GelMA group retained more within the construct as shown by immunohistochemistry. This is consistent with lap-shear data that showed the 16% GelMA group had a significantly lower kinetic coefficient of friction as compared to 14% GelMA/2% OMA. Theoretically, more lubricin was still present in the construct, effectively reducing the coefficient of friction. The lubricin immunohistochemical staining was rather sparse overall, and this could be due to lubricin also being secreted. It could also be due to the cell density within the construct. The cell density was kept consistent from the screen through the bioprinting, originally chosen to compare between the cell aggregate control and the biomaterials. However, the surface zone of articular cartilage has a higher cell density [55] than the deeper zones, and mimicking that in vitro may increase lubricin expression. It should also be noted that the chondrocytes used in this study were a mixed population, and thus not exclusively surface zone chondrocytes. It has been shown that even with TGFβ stimulation, middle and deep zone chondrocytes had barely any lubricin expression [3]. Incorporating solely surface zone chondrocytes may result in higher lubricin expression.

One of the requirements for an ideal tissue engineered articular cartilage construct is that it needs to not only fill the defect space but maintain that space while new tissue forms [19]. This means that the 3D bioprinted construct must retain its shape while forming cartilage tissue. The 3D bioprinted constructs with articular chondrocytes encapsulated had a noticeable decrease in size in both the 14% and 16% GelMA groups, while the 14% GelMA/2% OMA group retained its shape. Both the 14% and 16% GelMA groups had a significantly smaller surface area by day 22 as compared to 14% GelMA/2% OMA. The retention of area by 14% GelMA/2%OMA could be due to the higher GAG content and the increased distribution of type II collagen in the construct. Increased ECM could be replacing degrading biomaterial, though from the biochemical content assessed it is unlikely that this is completely replaced. This stabilizing effect is likely because of the presence of OMA and its controllable degradation rate. Because the OMA can be crosslinked with the GelMA, it is possible that the chondrocytes cannot biodegrade the gel. This would need to be assessed further. However, both GelMA and OMA have been shown to biodegrade individually [39,40,41,56,57,58]. OMA primarily biodegrades through hydrolysis, whereas GelMA primarily biodegrades through enzymatic degradation. We would expect the composite gel to degrade by the same mechanisms.

There is a chance that by adding 2% OMA to 16% GelMA we could see more lubricin secretion and construct stabilization over the 22 days; however, higher final percentages of biomaterials increase the difficultly in handling and runs the risk of further reducing cell viability. Finding the balance between biomaterial degradation and tissue formation will continue to be a challenge of tissue engineering, but biomaterials like OMA can facilitate finding that equilibrium. Overall, we have demonstrated the utility of extracellular matrix-driven secreted reporters in optimizing bioinks beyond the usual printability and cell viability metrics.

## 5. Conclusions

This study successfully determined an optimal combination of GelMA and OMA for surface zone articular cartilage 3D bioprinting focused on lubricin expression using our novel HuPRG4gLuc reporter cell system. Using the DoE in tandem with the HuPRG4gLuc cells created a more streamlined and systematic approach for testing biomaterials for 3D bioprinting. The 16% GelMA group had the highest luminescence, and it also retained more lubricin within the construct as compared to 14% GelMA/2% OMA. The 14% GelMA/2% OMA group had higher lubricin secretion, was easier to print with, and had better shape stability over the 22 days in culture. Together these results indicate the 14% GelMA/2% OMA after 15 s crosslinking as the optimal combination tested for lubricin expression and 3D bioprinting of surface zone articular cartilage.

## Figures and Tables

**Figure 1 bioengineering-10-00997-f001:**
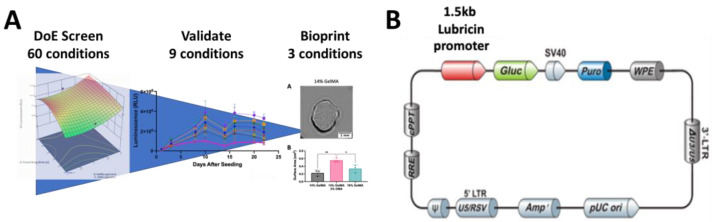
Overall project flow and plasmid map for PRG4gLuc. The overall project flow, from the DoE screen through the validation experiments to the final bioprinting conditions (**A**). Lubricin promoter-driven Gaussia luciferase (Gluc) plasmid map (9394 Bp, Genecopoeia). Contains puromycin (Puro) and ampicillin (Amp) selection cassettes (**B**).

**Figure 2 bioengineering-10-00997-f002:**
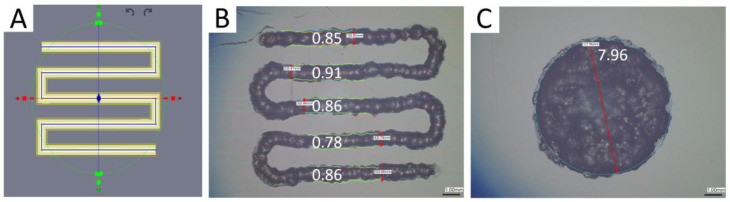
GelMA/OMA printability. A three-line print was designed in TSIM (**A**) which produced the lines shown (**B**) and the disc (**C**) using the settings in Table 1. All measurements are in mm. Scale bar 1mm.

**Figure 3 bioengineering-10-00997-f003:**
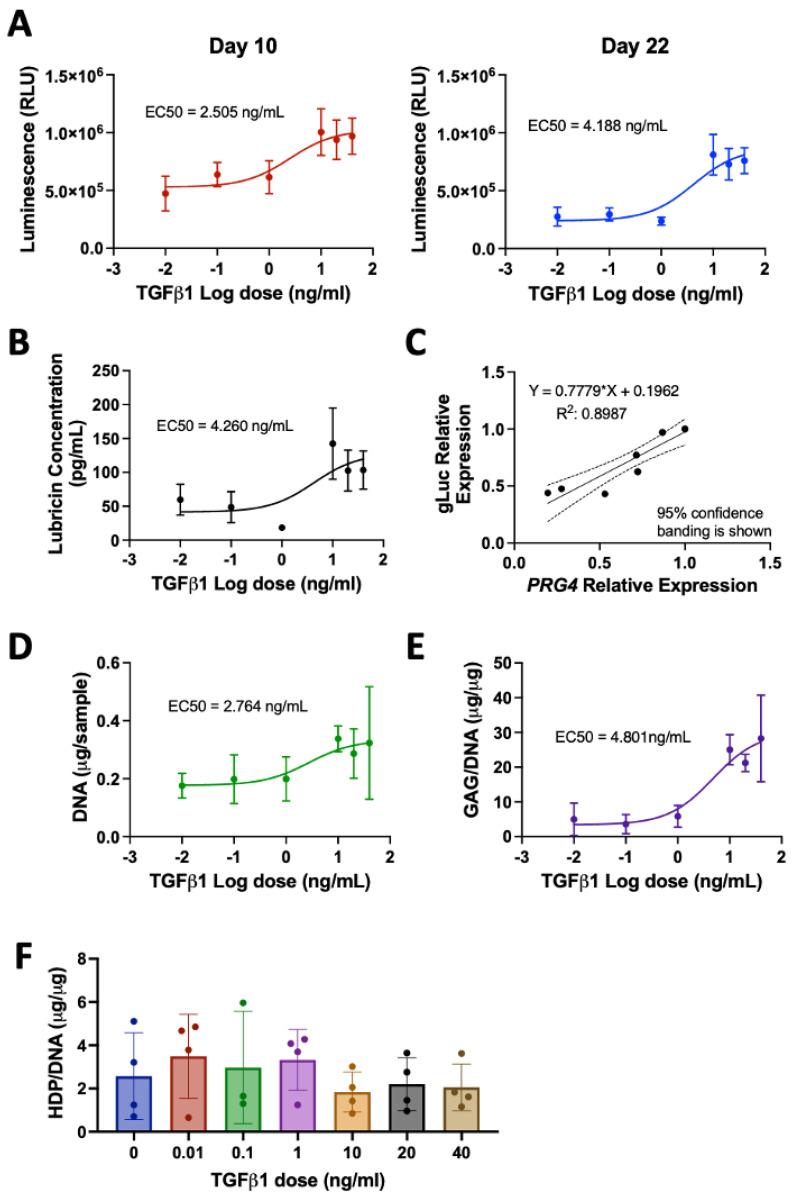
HuPRG4gLuc characterization. Primary human HuPRG4gLuc chondrocytes were grown in aggregate culture with TGFβ1 (0-40ng/mL). (**A**) Luminescence dose response curves for days 10 and 22. (**B**) Secreted lubricin concentration quantified by ELISA. (**C**) Relative gLuc gene expression correlates with relative PRG4 gene expression, with 95% confidence bands shown. (**D**) DNA (µg/sample) and (**E**) GAG/DNA (µg/µg) dose response curves. (**F**) HDP/DNA (µg/µg) at all concentrations of TGFβ1. n = 4–6, Ave. ± S.D.

**Figure 4 bioengineering-10-00997-f004:**
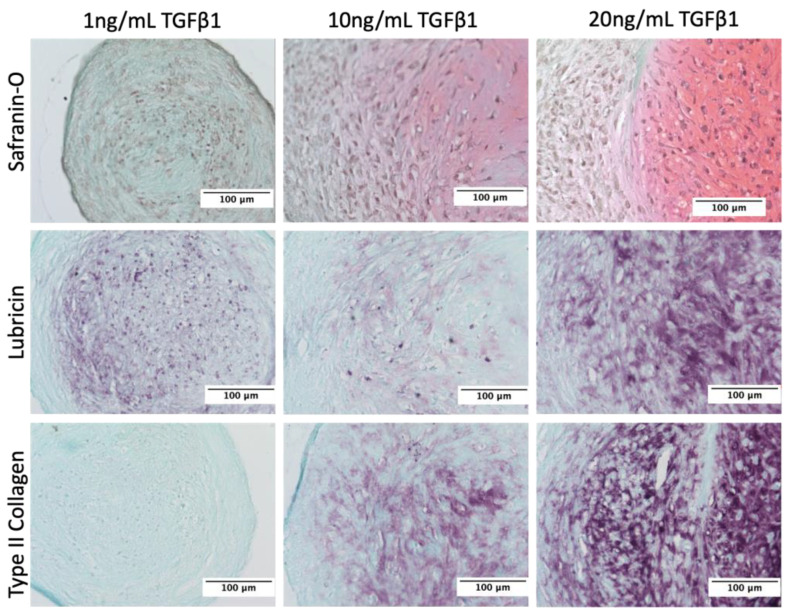
Histological analysis of cartilage aggregate response to TGFβ1. End of culture (day 22) histology staining of HuPRG4gLuc chondrocytes grown in aggregate culture with TGFβ1. Columns are 1, 10, and 20 ng/mL doses of TGFβ1. The top row is Safranin-O (red) staining for GAG content. The middle row is immunohistological staining (purple) for lubricin and the bottom row is for type II collagen. Scale bar shows 100 µm.

**Figure 5 bioengineering-10-00997-f005:**
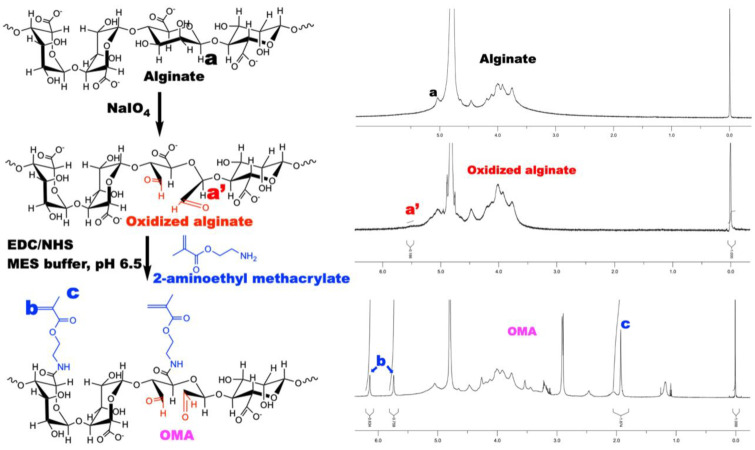
Biodegradable and photocrosslinkable OMA preparation and characterization. Sodium alginate was oxidized with sodium periodate in an aqueous solution, and then methacrylates were introduced onto the oxidized alginate. ^1^H-NMR spectra are shown with the top being alginate, the peak (a) being the protons on carbon 1; the middle showing oxidized alginate and the new protons formed on carbon 1 (a′); and the bottom for methacryloyl groups (b,c).

**Figure 6 bioengineering-10-00997-f006:**
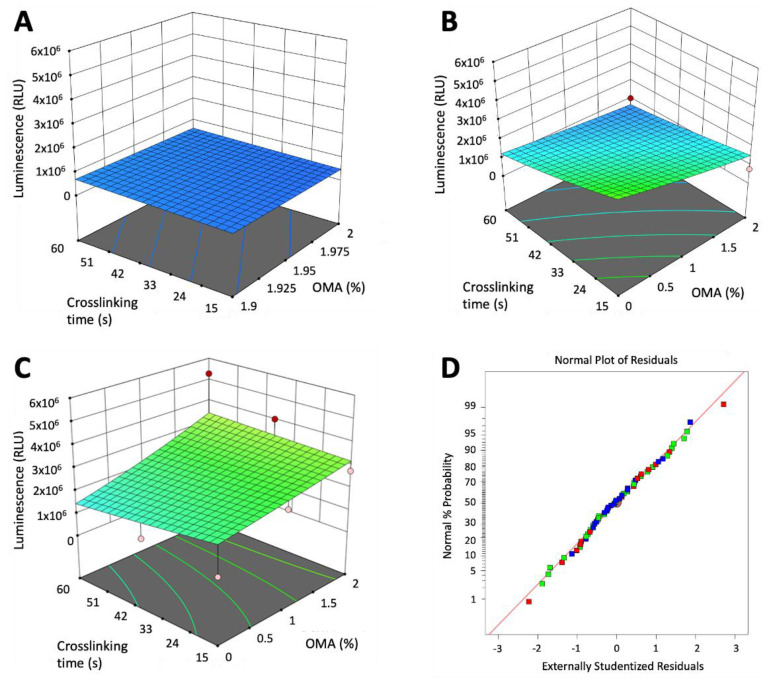
DoE identification of factors impacting lubricin expression. Design of experiment generated combinations of GelMA and OMA at different crosslinking times were mixed with HuPRG4gLuc cells and luminescence assessed over 22 days. (**A**) Groups with 0% GelMA, (**B**) 6% GelMA, and (**C**) 12% GelMA. (**D**) Corresponding normal probability plot of the residuals.

**Figure 7 bioengineering-10-00997-f007:**
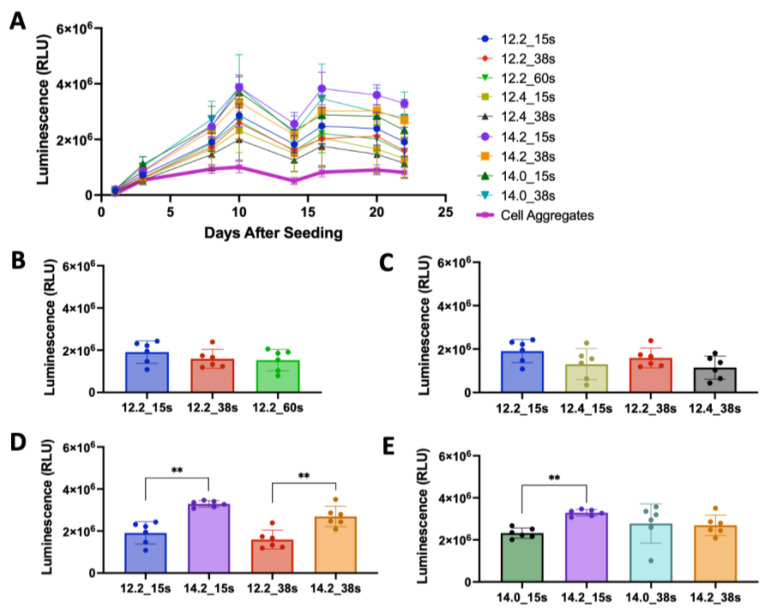
Validation of bioink combinations. (**A**) Biomaterial groups and cell aggregate control luminescence over 22 days. (**B**) 12% GelMA, 2% OMA luminescence with changing crosslinking time. (**C**) 12% GelMA with either 2% or 4% OMA after either 15s or 38s crosslinking. (**D**) 14% GelMA/2% OMA compared to 12% GelMA/2% OMA after 15s or 38s crosslinking. (**E**) 14% GelMA alone compared to 14% GelMA/2% OMA after 15s or 38s crosslinking. ** *p* value < 0.0001, n = 6, Ave. ± S.D. (Key: %GelMA: %OMA_crosslinking time(s)).

**Figure 8 bioengineering-10-00997-f008:**
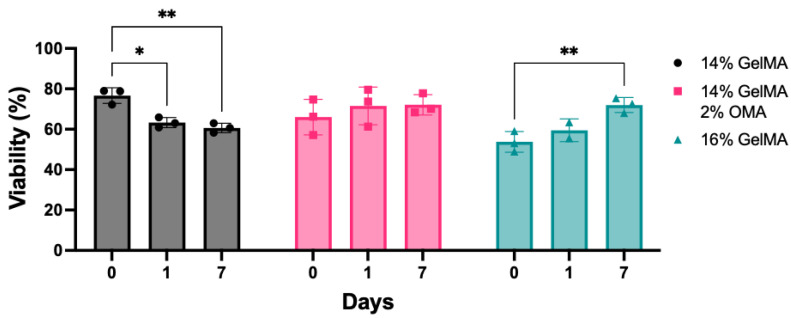
Cell viability of 3D bioprinted groups. Bioprinted 14% GelMA, 14% GelMA/2% OMA, and 16% GelMA on days 0, 1, and 7. * *p* value < 0.05, ** *p* value < 0.001, n = 3, Ave. ± S.D.

**Figure 9 bioengineering-10-00997-f009:**
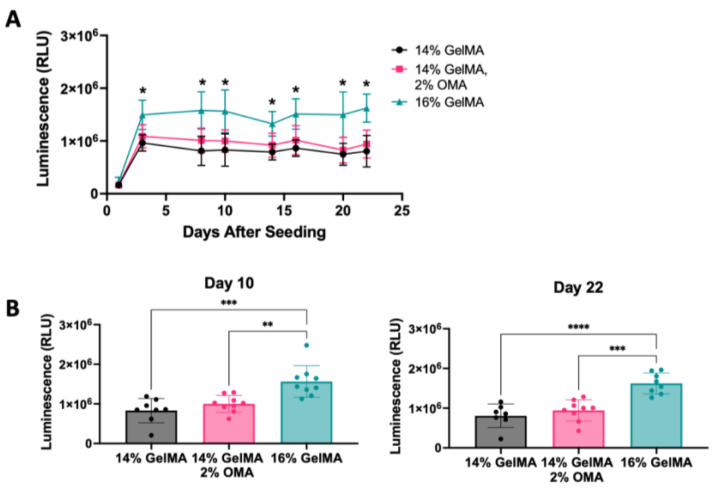
Bioprinting assessment of optimal bioink. Luminescence for 3D bioprinted groups 14% GelMA, 14% GelMA/2% OMA, and 16% GelMA. (**A**) Luminescence over 22 days in culture, 16% GelMA significance compared to the other groups (* *p* value < 0.001). (**B**) Day 10 and day 22 luminescence. **** *p* value < 0.0001, *** *p* value = 0.0001, ** *p* value < 0.005. n = 7–9, Ave. ± S.D.

**Figure 10 bioengineering-10-00997-f010:**
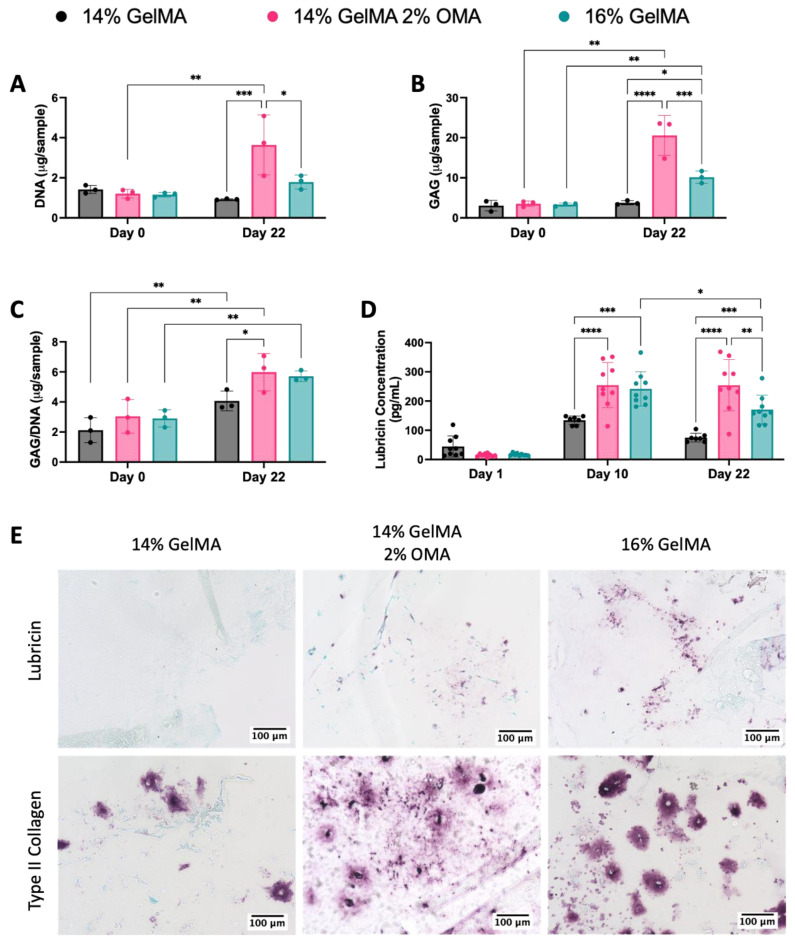
Biochemical and immunohistological characterization of 3D bioprinted constructs. (**A**) DNA content (µg/sample), (**B**) GAG content (µg/sample), and (**C**) GAG/DNA (µg/µg). (**D**) Secreted lubricin concentration (pg/mL) quantified by ELISA on days 1, 10, and 22. (**E**) Immunohistological staining of (top row) lubricin and (bottom row) type II collagen. 20× magnification with 100 µm scale bar. **** *p* value < 0.0001, *** *p* value = 0.0001, ** *p* value < 0.005, * *p* value < 0.05. n = 3–9, Ave. ± S.D.

**Figure 11 bioengineering-10-00997-f011:**
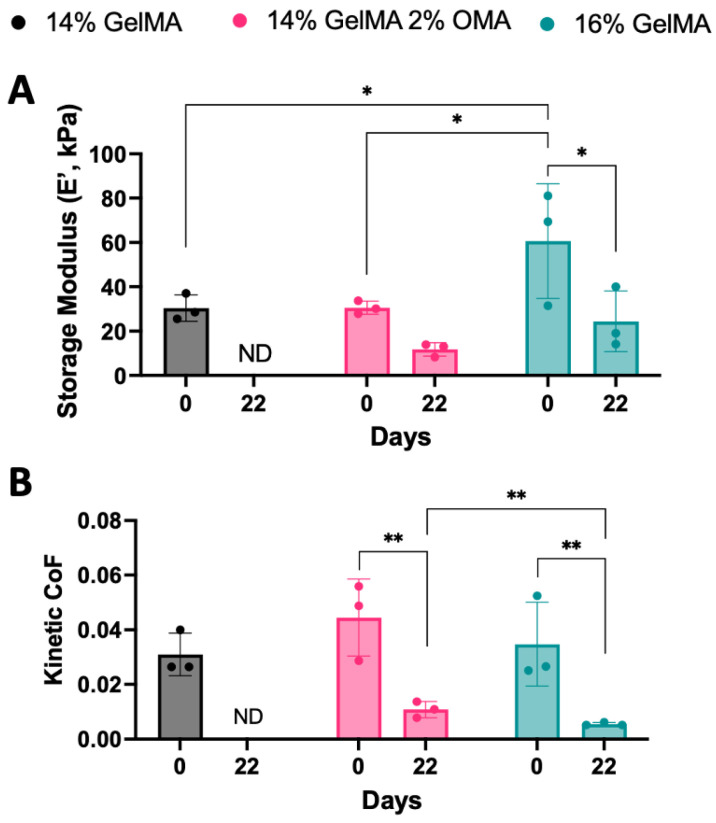
Mechanical characterization of 3D bioprinted groups. (**A**) Storage modulus determined by DMA (1 Hz). (**B**) Kinetic coefficient of friction determined by lap-shear testing. ** *p* value < 0.005 and * *p* value < 0.05. ND = not determined. n = 3, ± S.D.

**Figure 12 bioengineering-10-00997-f012:**
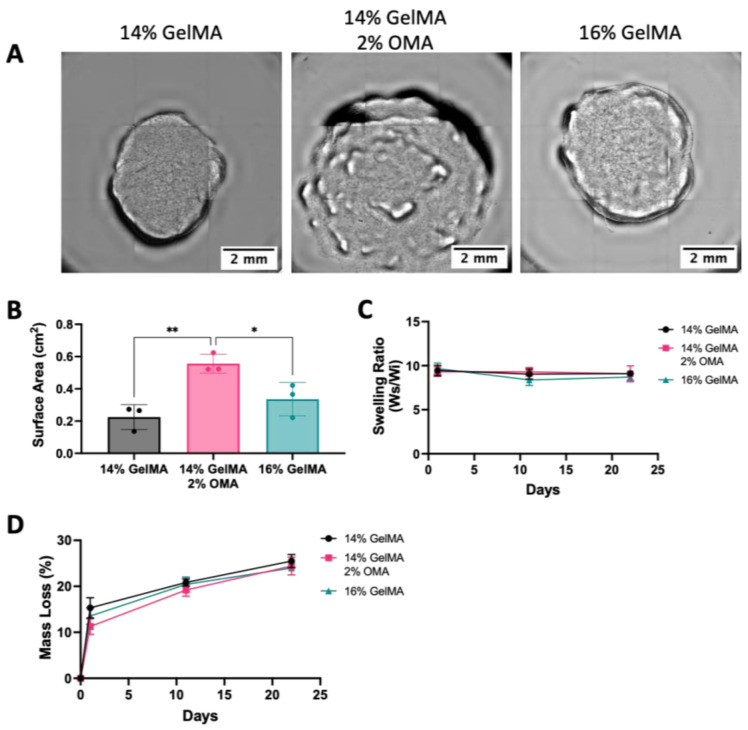
Shape fidelity and degradation. (**A**) Day 22 images of 14% GelMA, 14% GelMA/2%OMA, and 16% GelMA. 4× images with 2mm scale bar. (**B**) Surface area of day 22 constructs. (**C**) Swelling ratio over 22 days and (**D**) Mass loss percentage over 22 days. ** *p* value < 0.005 and * *p* value < 0.05. n = 3, ± S.D.

**Table 1 bioengineering-10-00997-t001:** 3D Bioprinting Settings.

Bioink	Pressure (psi)	Acceleration (mm/s^2^)	Speed (mm/s)	Ink Temp (°C)
14% GelMA	16–14	100	9	24
14% GelMA, 2% OMA	33–18	100	5	25
16% GelMA	15–12	100	7	25

## Data Availability

Data is contained within the article or Appendix A.

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
