# Peer review of "Optimizing Bioink Composition for Human Chondrocyte Expression of Lubricin"

_bioengineering, 2023, doi:10.3390/bioengineering10090997_

Round 1

Reviewer 1 Report

General comment:

The paper mainly presents two key contributions: 1. The new bioink based on gelatine methacrylate and oxidized methacrylated alginate for 3D bioprinting of artificial cartilage. 2. a novel reporter system with a PRG4 promoter-driven Gaussia luciferase (HuPRG4gLuc) in primary human chondrocytes.  The results are of interest in the area of additive manufacturing and bring a degree of innovation. The manuscript is well-written but requires some changes. The work may be considered for publication after major revision. Comments and suggestions are listed below.

Specific comments:

1.      The first concern is a lack of studies on bioink printability. The printability should be assessed quantitatively by e.g. calculating the difference between the actual printed line width and the nozzle diameter. It is highly important for printing precision. There are no images of any printouts, which could be confirmed that the bioink is printable.

2.      Could you please provide in SI the design of the experiment (DoE) matrix? What model do you refer to in section 2.7? What ranges of variables did you use?

3.      What is the optimal composition of the bioink? The surfaces are flat and show no local extremum, hence the optimum is at the edge of the variable range. This may suggest that the range has been incorrectly matched. Please expand the section.

4.      Could please provide full ANOVA results?

5.      Lines 644-645: “The results of this screen showed the highest final percentage of GelMA and OMA, 12% and 2% respectively, had the highest lubricin expression.” Could you discuss why and how this composition improves cell activity?

6.      Lines 679-680: “3D bioprinted constructs with articular chondrocytes encapsulated had a noticeable decrease in size in both the 14% and 16% GelMA groups, while the 14% GelMA/2% OMA group retained its shape.” Could you please show the constructs and confirm the shape fidelity?

Author Response

 Reviewer 1:
General comment:

The paper mainly presents two key contributions: 1. The new bioink based on gelatine methacrylate and oxidized methacrylated alginate for 3D bioprinting of artificial cartilage. 2. a novel reporter system with a PRG4 promoter-driven Gaussia luciferase (HuPRG4gLuc) in primary human chondrocytes.  The results are of interest in the area of additive manufacturing and bring a degree of innovation. The manuscript is well-written but requires some changes. The work may be considered for publication after major revision. Comments and suggestions are listed below.

Specific comments:

1.      The first concern is a lack of studies on bioink printability. The printability should be assessed quantitatively by e.g. calculating the difference between the actual printed line width and the nozzle diameter. It is highly important for printing precision. There are no images of any printouts, which could be confirmed that the bioink is printable.

These have now been included in the document (Fig. 2). There is also an image of the bioprinted discs in Figure 11A.

2.      Could you please provide in SI the design of the experiment (DoE) matrix? What model do you refer to in section 2.7? What ranges of variables did you use?

These have now been included in the SI (1S). The model was a quadratic by quadratic.

3.      What is the optimal composition of the bioink? The surfaces are flat and show no local extremum, hence the optimum is at the edge of the variable range. This may suggest that the range has been incorrectly matched. Please expand the section.

The optimal combination in our work was the 14% GelMA, 2% OMA which is stated in the conclusions. It is true that the range could be extended but with higher concentrations of GelMA and OMA we found it too difficult to work with. This has been added to the discussion.

4.      Could please provide full ANOVA results?

The details for the DoE ANOVAs have been included in the supplemental data (2S).

  1. Lines 644-645: “The results of this screen showed the highest final percentage of GelMA and OMA, 12% and 2% respectively, had the highest lubricin expression.” Could you discuss why and how this composition improves cell activity?

“This was thought to be a combination of the effect of hydrogel stiffness and composition, as was found with type II collagen expression” has been added to the discussion.

6.      Lines 679-680: “3D bioprinted constructs with articular chondrocytes encapsulated had a noticeable decrease in size in both the 14% and 16% GelMA groups, while the 14% GelMA/2% OMA group retained its shape.” Could you please show the constructs and confirm the shape fidelity?

The constructs are shown in Figure 11 part A and now, the original print shape in Fig. 2C.

Reviewer 2 Report

This study reported a novel chondrocyte reporter system using PRG4 promoter-driven Gaussia luciferase and optimized the GelMA/OMA bioink for 3D bioprinting. The whole experimental design is very comprehensive involving histological, mechanical, and biochemical evaluation. The following comments are provided for the authors’ consideration:

1.     In the abstract, “Therefore, 14% GelMA/2% OMA, 14% GelMA 20 and 16% GelMA were 3D bioprinted.”, as I read it is difficult to understand the logic here why these compositions were chosen. The authors should explain more clearly in the abstract and in the context.

2.     The authors are suggested to use an overall experimental scheme to illustrate the process of the whole study.

3.     An image to illustrate the experimental setup of mechanical lap-shear test is suggested to be added in the manuscript or supplementary info.

4.     In figure 4, the spectrum of alginate is suggested to be added.

Author Response

Reviewer 2:
This study reported a novel chondrocyte reporter system using PRG4 promoter-driven Gaussia luciferase and optimized the GelMA/OMA bioink for 3D bioprinting. The whole experimental design is very comprehensive involving histological, mechanical, and biochemical evaluation. The following comments are provided for the authors’ consideration:

1.     In the abstract, “Therefore, 14% GelMA/2% OMA, 14% GelMA 20 and 16% GelMA were 3D bioprinted.”, as I read it is difficult to understand the logic here why these compositions were chosen. The authors should explain more clearly in the abstract and in the context.

The abstract has been modified to “Therefore, DoE optimized 14% GelMA/2% OMA, 14% GelMA control and 16% GelMA (total solid content control) were 3D bioprinted.” The sentence in 3.6 has been modified to: “Controls: 14% GelMA, as a GelMA only control, and 16% GelMA, as a total solid content control, both with 15s crosslinking were also printed into discs.”

  1. The authors are suggested to use an overall experimental scheme to illustrate the process of the whole study.

An overall project flow has been included as part of figure 1.

  1. An image to illustrate the experimental setup of mechanical lap-shear test is suggested to be added in the manuscript or supplementary info.

This is included as a supplementary figure S2

4.     In figure 4, the spectrum of alginate is suggested to be added.

This has been added

Reviewer 3 Report

The authors attribute the usefulness of methacryloyl gelatin (GelMA) and oxidized methacrylated alginate (OMA) composites as bioinks that can induce lubricin expression in cartilage tissue. The paper concludes that a combination of 14% GelMA and 2% OMA is the optimal composition of bioinks that can maintain high lubricin expression and retention of 3D printed structures at reasonable levels. Although this paper suggests that mixed GelMA and OMA biomaterials may be very useful for cartilage tissue construction in the field of 3D bioprinting, I have some questions and comments as shown below:

The authors showed a lot of data at day 22 in various outcomes, but only up to day 7 for cell viability. Although they evaluated the amount of DNA at day 22, I recommend that the authors evaluate cell viability at day 22 to be consistent with other data. (in Figure 7)

In this study, the authors used the mixture of GelMA and OMA as biodegradable materials, but the swelling and degradability of the mixed hydrogel without cells were not significantly different from those of GelMA alone from Figures 11c-d. On the other hand, when cells were mixed, the surface areas of the gel disks were significantly smaller for GelMA alone, whereas the GelMA+OMA gels caused almost no reduction in surface area. The authors speculate that the increase in ECM such as GAGs may be replaced by degraded biomaterials. Considering the original volume of the disc and the concentrations of GelMA and OMA, it can be estimated that the material mass of the bioink is approximately 8 mg, while the increase in GAG is only ~50 μg at most. It is unlikely that this amount of GAG increase would be sufficient to maintain the gel structure. Therefore, it is more reasonable to assume that the presence of OMA contributes to the stability of the gel structure, as the authors concluded in the text that immediately follows. If so, it is highly questionable whether this bioink, a mixture of GelMA and OMA, can be said to be biodegradable. Based on their results, it is more natural to assume that the bioink used in this paper is probably not biodegradable for chondrocytes. The authors need to discuss this point more carefully and cautiously.

Author Response

 Reviewer 3:
The authors attribute the usefulness of methacryloyl gelatin (GelMA) and oxidized methacrylated alginate (OMA) composites as bioinks that can induce lubricin expression in cartilage tissue. The paper concludes that a combination of 14% GelMA and 2% OMA is the optimal composition of bioinks that can maintain high lubricin expression and retention of 3D printed structures at reasonable levels. Although this paper suggests that mixed GelMA and OMA biomaterials may be very useful for cartilage tissue construction in the field of 3D bioprinting, I have some questions and comments as shown below:

The authors showed a lot of data at day 22 in various outcomes, but only up to day 7 for cell viability. Although they evaluated the amount of DNA at day 22, I recommend that the authors evaluate cell viability at day 22 to be consistent with other data. (in Figure 7)

Cell viability by fluorescence measurement becomes more difficult as cells produce extracellular matrix which is autofluorescent and, by day 22, is a combination of viability and proliferation. This viability is reflected in the DNA assay (Fig. 9A).

In this study, the authors used the mixture of GelMA and OMA as biodegradable materials, but the swelling and degradability of the mixed hydrogel without cells were not significantly different from those of GelMA alone from Figures 11c-d. On the other hand, when cells were mixed, the surface areas of the gel disks were significantly smaller for GelMA alone, whereas the GelMA+OMA gels caused almost no reduction in surface area. The authors speculate that the increase in ECM such as GAGs may be replaced by degraded biomaterials. Considering the original volume of the disc and the concentrations of GelMA and OMA, it can be estimated that the material mass of the bioink is approximately 8 mg, while the increase in GAG is only ~50 μg at most. It is unlikely that this amount of GAG increase would be sufficient to maintain the gel structure. Therefore, it is more reasonable to assume that the presence of OMA contributes to the stability of the gel structure, as the authors concluded in the text that immediately follows. If so, it is highly questionable whether this bioink, a mixture of GelMA and OMA, can be said to be biodegradable. Based on their results, it is more natural to assume that the bioink used in this paper is probably not biodegradable for chondrocytes. The authors need to discuss this point more carefully and cautiously.

This is a valid point and has been discussed more. We did not demonstrate biodegradability of the hydrogels (lines 690-4).

Round 2

Reviewer 1 Report

The paper now can be accepted for publication in Bioengineering.

Author Response

Many thanks for your comment.

The following sentences have been added to the discussion “However, both GelMA and OMA have been shown to biodegrade individually [39,40,56–58]. OMA primarily biodegrades through hydrolysis, whereas GelMA primarily biodegrades through enzymatic degradation [57,59].”

I hope that is satisfactory.

Best regards
T. Kean